# Neurons Co-Expressing GLP-1, CCK, and PYY Receptors Particularly in Right Nodose Ganglion and Innervating Entire GI Tract in Mice

**DOI:** 10.3390/ijms26052053

**Published:** 2025-02-26

**Authors:** Elizabeth Laura Lansbury, Vasiliki Vana, Mari Lilith Lund, Mette Q. Ludwig, Esmira Mamedova, Laurent Gautron, Myrtha Arnold, Kristoffer Lihme Egerod, Rune Ehrenreich Kuhre, Jens Juul Holst, Jens Rekling, Thue W. Schwartz, Stanislava Pankratova, Oksana Dmytriyeva

**Affiliations:** 1Novo Nordisk Foundation Center for Basic Metabolic Research, Faculty of Health and Medical Sciences, University of Copenhagen, DK-2100 Copenhagen, Denmark; elizabeth.lansbury@sund.ku.dk (E.L.L.); vsvn@novonordisk.com (V.V.); mari.l.rasmussen@gmail.com (M.L.L.); metteludwig@hotmail.com (M.Q.L.); egerod@sund.ku.dk (K.L.E.); ruku@novonordisk.com (R.E.K.); jjholst@sund.ku.dk (J.J.H.); tws@sund.ku.dk (T.W.S.); stasya@sund.ku.dk (S.P.); 2Institute of Neuroscience, Faculty of Health and Medical Sciences, University of Copenhagen, DK-2200 Copenhagen, Denmark; esmira@sund.ku.dk (E.M.); jrekling@sund.ku.dk (J.R.); 3Center for Hypothalamic Research, Department of Internal Medicine, The University of Texas Southwestern Medical Center, Dallas, TX 75390, USA; laurent.gautron@utsouthwestern.edu; 4Department of Health Sciences and Technology, ETH Zurich, 8092 Zurich, Switzerland; myrtha-arnold@ethz.ch; 5Department for Biomedical Research, Faculty of Health Sciences, University of Copenhagen, DK-2200 Copenhagen, Denmark; 6Department of Veterinary and Animal Science, Faculty of Health and Medical Sciences, University of Copenhagen, DK-1870 Copenhagen, Denmark

**Keywords:** nodose ganglia, vagal afferent neurons, GLP-1 receptor, CCKa receptor, NPY2 receptor, neurotensin receptor

## Abstract

Afferent vagal neurons convey gut–brain signals related to the mechanical and chemical sensing of nutrients, with the latter also mediated by gut hormones secreted from enteroendocrine cells. Cell bodies of these neurons are located in the nodose ganglia (NG), with the right NG playing a key role in metabolic regulation. Notably, glucagon-like peptide-1 receptor (GLP1R) neurons primarily innervate the muscle layer of the stomach, distant from glucagon-like peptide-1 (GLP-1)-secreting gut cells. However, the co-expression of gut hormone receptors in these NG neurons remains unclear. Using RNAscope combined with immunohistochemistry, we confirmed GLP1R expression in a large population of NG neurons, with *Glp1r*, *cholecystokinin A receptor (Cckar)*, and *Neuropeptide Y Y2 Receptor (Npy2r)* being more highly expressed in the right NG, while *neurotensin receptor 1 (Ntsr)*, G protein-coupled receptor (*Gpr65*), and *5-hydroxytryptamine receptor 3A (5ht3a)* showed equal expressions in the left and right NG. Co-expression analysis demonstrated the following: (i) most *Glp1r*, *Cckar*, and *Npy2r* neurons co-expressed all three receptors; (ii) nearly all *Ntsr1*- and *Gpr65*-positive neurons co-expressed both receptors; and (iii) *5ht3a* was expressed in subpopulations of all peptide-hormone-receptor-positive neurons. Retrograde labeling demonstrated that the anterior part of the stomach was preferentially innervated by the left NG, while the right NG innervated the posterior part. The entire gastrointestinal (GI) tract, including the distal colon, was strongly innervated by NG neurons. Most importantly, dual retrograde labeling with two distinct tracers identified a population of neurons co-expressing *Glp1r*, *Cckar*, and *Npy2r* that innervated both the stomach and the colon. Thus, neurons co-expressing GLP-1, cholecystokinin (CCK), and peptide YY (PYY) receptors, predominantly found in the right NG, sample chemical, nutrient-induced signals along the entire GI tract and likely integrate these with mechanical signals from the stomach.

## 1. Introduction

The vagus is the Xth cranial nerve, which senses and provides the parasympathetic control of nearly all visceral organs of the thoracic and abdominal cavities, including the heart, lungs, liver, pancreas, and gastrointestinal (GI) tract. The two-way neuronal signaling of the vagal nerve occurs through approximately 90% afferent sensory and around 10% efferent motor fibers [1]. The cell bodies of the vagal motor neurons are located in the dorsal motor nucleus of the vagus, which is part of the dorsal vagal complex (DVC) in the brainstem. In contrast, the pseudo-bipolar cell bodies of the afferent sensory vagal neurons reside in the right and left nodose ganglia (NG), situated just below the site where the vagal nerve exits/enters the cranium. Each of these NG neurons send their axons up to neurons in the corresponding left and right nucleus tractus solitarius in the DVC and extend their long sensory dendrite(s) down to monitor the status of different visceral organs [2,3]. For the GI tract, the vagus nerve constitutes the main pathway for gut–brain signaling. It senses the presence of dietary nutrients through both the mechano-sensation of gastric and intestinal distention and the chemo-sensation of nutrients, at least partly mediated by gut hormones released from enteroendocrine cells. These hormones, including 5-hydroxytryptamine (5-HT), are directly sensed by specific receptors expressed on afferent vagal neurons. Although peptide gut hormones such as CCK, GLP-1, PYY, and neurotensin also function as circulating endocrine messengers, i.e., hormones, their signaling though afferent vagal neurons is a major component of their physiological gut–brain signaling mechanism, particularly for CCK and GLP-1 [4,5,6,7,8,9,10]. Thus, it is thought that afferent vagal nerve fibers innervate the submucosa of the intestinal villi, where they sense the gut hormones secreted in response to dietary nutrients through receptors expressed on their nerve endings. It has even been proposed that synapses are formed between enteroendocrine cells and afferent vagal nerve endings [11,12]. However, in 2016, Liberles and coworkers introduced a whole new concept for understanding the afferent vagal sensing and signaling from the GI tract. By the use of genetically guided mapping, functional studies with in vivo single-neuron imaging, and optogenetics, they described that GPR65-expressing neurons, rather than GLP-1-receptor-expressing neurons, innervate the intestinal villi to detect nutrients and functionally control gut motility [13]. In contrast, surprisingly, GLP1R-expressing neurons mainly innervate the muscle layer of the stomach and, to lesser extent, the intestine, forming specialized terminals that detect stretch rather than nutrients [13]. Three years later, Knight and coworkers expanded on this concept by providing a detailed overview of vagal sensory neurons based on single-cell transcriptomic approaches applied to several genetic models combined with advanced functional imaging studies [14]. The highlight of this granulated picture of vagal afferent sensing was—in agreement with the conclusions of Liberles and coworkers—that afferent vagal intra-ganglionic laminar endings (IGLEs) expressing GLP-1 receptors primarily sense mechanical stretch in the stomach and duodenum, inhibiting food intake, rather than sensing nutrients [13]. This is not the case for the NG neurons innervating the intestinal mucosa, which express GPR65 among many other genes.

An often overlooked feature of the vagal system is its asymmetrical organization into left and right vagal nerves, each with a corresponding NG, which, at least in the thoracic cavity, innervate different organs, such as the left lung versus right lung, respectively [1,15,16]. Although abdominal vagal innervation is also asymmetric, this is generally not appreciated, particularly in relation to the GI tract, except for the different innervations of the anterior and posterior parts of the stomach, which have distinct embryological origins [17]. However, recently, de Araujo and coworkers described that afferent vagal neurons are essential components of a reward neuronal pathway linking sensory neurons in the upper gut to dopamine release in the striatum. Specifically, they showed that the gut hormone CCK’s control of food intake is primarily mediated through the right NG [6]. This indicates that, at least for CCK, gut–brain signaling occurs asymmetrically, mainly via the right NG. Whether this is also the case for other gut hormones is unknown. Although several single-cell RNAseq studies of NG neurons have been published, none have differentiated between left and right NG [4,18]. Moreover, gut hormone G protein-coupled receptor (GPCR) sensors, like many other GPCRs, are generally expressed at rather low mRNA levels, making RNAseq a suboptimal method for quantifying their expression.

In the present study, we used the highly sensitive RNAscope in situ hybridization technology, designed to detect and visualize mRNA in individual cells, to characterize the expression and co-expression of GPCR sensors for gut hormones in both the left and the right NG, separately. We studied the expression of receptors for GLP-1 (*Glp1r*), CCK (*Cckar*), PYY (*Npy2r*), and neurotensin (*Ntsr1*), as well as GPR65, a marker of vagal neurons innervating intestinal mucosa, and the *5ht3a* subunit of the 5-HT/serotonin-gated ion channel receptor. Our results indicated that not only CCK, but also GLP-1 and PYY, in contrast to neurotensin and 5HT, primarily signal through the right NG. Furthermore, our retrograde tracings demonstrated that a significant population of afferent vagal neurons co-expressing receptors for CCK, GLP-1, and PYY innervate the entire GI tract, from the stomach to the colon. These preliminary findings could potentially explain the enigmatic presence of neurons apparently expressing GLP-1 receptors in the muscle layer of the stomach, where GLP-1 is not produced.

## 2. Results

### 2.1. Asymmetrical Expression of Glp1r in the Left and Right Nodose Ganglia

The total number of neurons was quantified separately in the left and right NG of wild-type mice using an unbiased stereological approach. Approximately 6000 neurons were counted in each ganglion, indicating an equal total cell number between the left and right NG (n = 7 mice per left/right ganglion; Figure 1a). We further extended our previous findings accomplished in whole NG [4] by examining the expression of *Glp1r* separately in the left and right NG of wild-type mice using qPCR. Surprisingly, the expression of *Glp1r* was significantly higher in the right ganglion (*p* < 0.01, n = 5 per left/right ganglion; Figure 1b). This asymmetric expression pattern was further substantiated by immunohistochemistry (IHC) using NG specimens from Glp1r-Cre transgenic mice, in which a fluorescent reporter was introduced into *Glp1r*-expressing cells through crossing with a tdTomato-flox reporter strain. As shown in Figure 1c, a large population of intensely red fluorescent Glp1r-tdTomato+-cells was co-labeled with the pan-neuronal marker HuC/HuD in NG from the reporter mice. The quantification of Glp1r-tdTomato+ neurons demonstrated that more than 50% of all neurons were Glp1r-tdTomato-positive in the right NG, whereas in the left ganglion, only ~30% of detected neurons were positive for the Glp1r reporter, i.e., a ~1.6 fold difference was detected (*p* < 0.0001; Figure 1d), which corresponds closely to the difference observed with qPCR (Figure 1b). Due to the construction of the transgenic reporter mouse strain, mature neurons could give rise to a false positive signal traced from immature stages, since *Glp1r* expression in vagal sensory neurons is known to rise significantly in the perinatal period [19]. To validate that *TdTomato*-expressing neurons also express *Glp1r* mRNA in adulthood, we performed a chromogenic Duplex RNAscope analysis, which revealed that over 90% of all TdTomato+ cells in the NG expressed mRNA, as shown in Figure 1 (Figure 1e).

GLP1R is one of the few GPCRs for which it is possible to determine its expression at the protein level, owing to the availability of a validated antibody. Using this monoclonal GLP1R antibody, we demonstrated GLP1R protein expression in neurons in both the left and right NG, with a ~1.7-fold higher number of GLP1R+ neurons observed in the right NG compared to the left (*p* < 0.001; Figure 1f,g). However, the number of GLP1R+ neurons in both the right and left NG was from approximately half to one-third of the count observed with the Glp1r-tdTomato reporter (Figure 1c,d) and the RNAscope analysis (Figure 1e).

We concluded that *Glp1r* is expressed in approximately 50% of the 6000 neurons in the right NG and approximately 30% of the neurons of the left NG in mice.

### 2.2. Similar Lateralization of Cckar and Npy2r but Not Ntsr1, Gpr65, and 5ht3a in Right Versus Left NG

Given the observed asymmetrical distribution of *Glp1r*, we wondered to what degree this phenomenon extended to the other receptors involved in sensing gut hormones and related signals. CCK is one such hormone whose action has been robustly linked to the activation of afferent vagal neurons [10,20]. The quantification of RNAscope-labeled *Cckar*+-cells revealed a pronounced asymmetrical distribution, with ~20% of *Cckar*+-neurons in the left and ~35% in the right NG (*p* < 0.05; Figure 2a,b), mirroring a very similar relative difference as that observed for *Glp1r* (Figure 1). This lateralization of Cckar expression was further validated by a qPCR analysis of mRNA isolated from the left and right NG, respectively (*p* < 0.001; Figure 2c). A comparable lateralization pattern was also observed for *Npy2r*, which encodes the Y2 receptor for Neuropeptide Y (NPY), which has a high affinity for the gut hormone PYY, co-expressed and co-released with GLP-1 [21]. This was determined both by RNAscope analysis (*p* < 0.01; Figure 2d,e) and qPCR (*p* < 0.01; Figure 2f). However, for *Npy2r*, the right-to-left ratio was smaller, probably due to the fact that a relatively large population of *Npy2r*-expressing neurons innervate the lungs but not the gut [22].

In contrast to *Glp1r*, *Cckar*, and *Npy2r*, the receptor for neurotensin (NTS), a gut peptide co-expressed and co-released, to some extent, with GLP-1 and PYY [23], *Ntsr1*, showed no lateralization. It was expressed in a similar number of neurons, approximately 17% in both the left and right NG (Figure 2g,h), and this symmetrical distribution was confirmed by qPCR (Figure 2i).

The orphan receptor Gpr65 has been identified as a marker of vagal neurons specifically innervating the mucosa of the intestine [13]. Similar to *Ntsr1*, RNAscope analysis revealed an equal distribution of *Gpr65*-expressing neurons between the left and right NG, accounting for ~15% of the total neuronal population in both ganglia (Figure 2j,k), which was confirmed by qPCR (Figure 2l).

5HT, released from enterochromaffin cells of the GI tract, is sensed by the homopentameric ligand-gated ion channel 5HT3a in afferent vagal neurons. As determined by RNAscope, nearly 50% of the neuronal cell bodies in both the left and right NG expressed *5ht3α* (Figure 2m,n), and this equal distribution was again confirmed by qPCR (Figure 2o).

### 2.3. Mapping of Co-Expression of Key Receptors in NG Neurons by RNAscope

Mapping the receptor expression in different subtypes or clusters of NG neurons, as identified by scRNA-seq analysis, is hampered by the fact that GPCRs are generally expressed at relatively low levels and, therefore, are frequently not picked up despite the fact that they may actually be expressed. To overcome this limitation, we applied a more sensitive RNAscope method here to investigate the co-expression of key receptors using triplex fluorescent RNAscope with probes targeting *Glp1r*, *Cckar*, *Npy2r*, *Ntsr1*, *Gpr65*, and *5ht3a* receptors in relevant combinations in both the right and left NG.

While the number of neurons expressing *Glp1r* differed between the left and right NG, as described earlier (Figure 1), the co-expression pattern of *Glp1r* with other receptors, expressed as a percentage of the total *Glp1r*-expressing cells, was remarkably similar in both sides of the NG (Appendix A). Thus, no apparent lateralization was observed for any of the observed co-expression patterns, except for a seemingly higher occurrence of otherwise relatively rare *Ntsr1+5ht3a+Glp1r* co-expressing neurons in the left compared to the right NG (Appendix A). Consequently, we analyzed the co-localization of these receptors in the combined left/right NG.

Initially, we focused on the potential co-localization of *Glp1r*, *Cckar*, and *Npy2r*, all showing significant lateralization within the NG (Figure 1 and Figure 2). We observed a high degree of co-expression among these three gut peptide receptors (Figure 3, panel a). Most notably, over 50% of all *Glp1r*+-neurons also expressed both *Cckar* and *Npy2r*, representing the co-expression of all three peptide receptors. Concerning *Cckar*+-neurons, 64% also expressed *Npy2r*, and 50% of these additionally expressed *Glp1r* (Figure 3a,d). However, a sizeable fraction of *Glp1r*+-neurons (37%) neither expressed *Cckar* nor *Npy2r*, likely constituting another distinct cluster of NG neurons (Figure 3a,d). Similarly, 50% of all *Npy2r*+-neurons were negative for the two other receptors, *Cckar* and *Glp1r* (Figure 3a,d), likely corresponding to the large population of *Npy2r*+-vagal neurons known to innervate the lungs [22,24].

The triplex RNAscope analysis of *Glp1r*+ together with *Gpr65* and *Ntsr1* revealed that less than 1% of *Glp1r*+-neurons co-expressed *Gpr65* and *Ntsr1* (Figure 3, panel b). This finding is consistent with previous observations indicating mutually exclusive expressions of *Glp1r* and *Gpr65* in nodose neurons [13]. Importantly, we unexpectedly discovered that the vast majority (86%) of *Gpr65*+ NG neurons, previously described to specifically innervate the intestinal mucosa and respond to nutrients and serotonin stimulation [13], also co-expressed *Ntsr1* (Figure 3, panel b).

Next, we analyzed the co-localization of *Glp1r* with *Ntsr1* and *5ht3α* (Figure 3, panel c), revealing that large fractions, 69% of *Ntsr1*+-neurons and 42% of *Glp1r*+-neurons, also expressed *5ht3α*. The mRNA encoding the *5ht3α* receptor was highly abundant in the NG and a considerable population (60%) of *5ht3α*+-neurons did not overlap with neither *Glp1r*- nor *Ntsr1*-expressing cells (Figure 3c,f). In line with this, the independent quantification of *5ht3α*+-cells co-expressing the above-mentioned receptors in the whole NG using the Duplex RNAscope showed that only 33% and 36% of *5ht3α*+-cells co-expressed *Glp1r* and *Ntsr1*, respectively (Appendix A). Of note, about half of *5ht3α*+-cells (48%) also co-expressed the orphan receptor *Gpr65* and a smaller fraction of *5ht3α*+-cells co-expressed *Cckar* (37%) and *Npy2r* (38%) (Appendix A).

Altogether, these data indicate that *Glp1r*+, *Cckar*+, and *Npy2r*+ co-expressing cells appear to constitute a sub-population of vagal neurons which are distinct from *Gpr65* and *Ntsr1* co-expressing cells, and, moreover, a relatively large fraction of all peptide-hormone-receptor-expressing NG neurons also express the *5ht3a* receptor.

### 2.4. Revisiting Glp1r Expression in NG Cell Clusters Based on Single-Cell RNAseq

Recent studies using sequencing analysis and complementary RNA-based techniques have highlighted the diversity of afferent neurons expressing different combinations of nutritional hormone-sensing receptors, notably including *Glp1r* [4,14]. However, in the recent ‘Atlas of Vagal Sensory Neurons’, *Glp1r*-expressing neurons were, surprisingly, missing [24]. Upon analysis of the raw data provided by Kupari and colleagues, we realized that the *Glp1r* transcript was, in fact, absent from their atlas due to a misannotation of the gene. Thus, we re-annotated the transcription end coordinate for *Glp1r* to target the most downstream polyadenylation site and adjusted the counting for *Glp1r* accordingly (see methods for details). Following these corrections, the expression of *Glp1r* could be profiled in neurons from the murine NG. In Appendix A, the *Glp1r* expression level and its distribution across the eighteen NG clusters identified by Kupari and coworkers are depicted in a violin plot [24]. A high expression of *Glp1r* was detected in neurons from clusters NG8, NG10, NG13, and, to a lesser extent, NG15, while lower levels of *Glp1r* expression were found in other clusters such as NG6 and NG16–NG18. Among these clusters, NG13 was also reported to express both *Cckar* and *Npyr2*, as well as *5ht3a*, and could consequently partly correspond to the NG neurons identified here by RNAscope to co-express these receptors (Figure 3). Interestingly, the stretch sensor *transient receptor potential cation channel subfamily V member 1 (Trpv1)* was also highly expressed in cluster NG13.

### 2.5. Retrograde Tracing Demonstrates NG Lateralization of Stomach Innervation and Strong Innervation of the Distal Intestine in Mice

To link NG lateralization to the vagal innervation of the GI tract, we injected the retrograde neuronal tracer, wheat germ agglutinin (WGA), in different locations of the GI tract in the rostro-caudal direction and quantified tracer-positive neurons in the left and right NG separately (Figure 4a,b). The retrograde labeling revealed that both the left and right ganglia innervate the GI tract in a region-specific manner, but with more tracer-positive cells receiving signals from the stomach, the pyloric region, and the duodenum (Figure 4b). The density of tracer-positive cells gradually decreased from the proximal to the distal small intestines, followed by an increase in the density of NG neurons innervating the caecum, the proximal, and even the distal colon (Figure 4b). Significant differences between the left and right NG were observed only for the afferents innervating the anterior and posterior parts of the stomach, where the left NG preferentially innervated the anterior part and the right NG preferentially innervated the posterior part of the stomach (Figure 4b). Surprisingly, the tracer injected in the distal parts of the gut, specifically the proximal and distal colon, was clearly detected in a substantial population of NG neurons. This finding indicates the robust innervation of the large intestine by vagal afferent sensory fibers, corroborating recently published data [25]. Interestingly, Wang and Taché reported that, in male mice, the right nodose ganglion contains a higher number of neurons innervating the colon than the left ganglion [26]. Our findings align with this observation, showing a similar trend, with tracer-positive neurons from colon injections being more prevalent in the right NG than in the left.

Multiplex RNAscope analysis applied to tracer-labeled NG demonstrated that the majority of tracer-positive neurons innervating the colon expressed either *Cckar* or co-expressed *Ntsr* and *Gpr65* (Figure 4c). Given that CCK is predominantly produced by enteroendocrine cells (EECs) of the proximal small intestines, we hypothesized that some nodose neurons may innervate multiple sites along the GI tract and detect different signals. To test this, we performed dual injections of retrograde tracers, each conjugated with different fluorophores, where one was injected into the stomach and the other into the colon, to assess whether individual neurons simultaneously innervate these two segments. As shown in Figure 5, we observed the retrograde transport of both tracers back to the same neurons within the NG, representing 23% of all analyzed NG neurons (Figure 5a, right). Importantly, by combining two different retrograde tracers with RNAscope-based analysis of peptide hormone receptor expression, we found that certain neurons innervating both the stomach and the colon also co-express *Cckar*, *Npy2r*, and *Glp1r* (Figure 5b,c). Furthermore, some *Cckar*+ and *Glp1r*+ NG neurons innervating both gut regions also express *Trpv1* and *Piezo2*, i.e., a polymodal nociceptor and a mechano-receptor [27,28], respectively (Figure 5d,e).

### 2.6. Functional Studies of NG Lateralization

To investigate whether receptor lateralization could be discerned in functional studies, we examined whole NG explants ex vivo [29], focusing on two extensively studied receptors, CCKAR and GLP1R, both of which exhibit pronounced lateralization in terms of NG expression. Bath application of CCK-8 (20 nM) induced similar electrophysiological responses in both the left and right ganglia, regardless of the ganglion type (n = 6 per left/right ganglia; Appendix A). Furthermore, consistent with the classical Gq-coupled nature of CCKAR, the application of 10 nM CCK-8 to NG explants transduced with the GECI Ca^2+^ censor resulted in elevated cytosolic Ca^2+^ levels in both the left and right NG (Appendix A). However, no apparent difference was observed between the left and right ganglia (Appendix A). In contrast, bath application of GLP-1 (10–100 nM) or equimolar amounts of exendin-4, a highly selective GLP1R agonist, induced no response in either Ca^2+^ imaging or electrophysiological nerve recordings (Appendix A), consistent with previous in vivo ganglion imaging following stimulation with agonists for the Gs-coupled GLP1R [13]. Overall, the left and right NG explants displayed a symmetrical responsiveness to CCK-8, but did not respond in terms of Ca^2+^ levels or electrophysiology to agonists for the Gs-coupled GLP1R alone.

## 3. Discussion

Recent studies employing highly advanced methods such as genetically guided mapping, single-cell transcriptomics applied to arrays of genetic models, and functional studies with in vivo single-neuron imaging have generated an enormous amount of data, providing a detailed picture of the complexity of afferent vagal innervation of the GI tract [30]. However, these studies have also left some intriguing unanswered questions. For instance, why do the afferent vagal neurons that innervate the mucosa of the small intestine and sense food to control gut motility not express receptors for gut hormones, but instead express the orphan GPR65 receptor [13,14]? Additionally, why do afferent vagal neurons that express GLP-1R primarily innervate the muscle layer of the stomach, where no GLP-1 is produced, and instead respond to distention to control food intake?

We believe that our relatively simple RNAscope analysis, which focuses on the expression and, importantly, the co-expression of gut hormone receptors, combined with the dual retrograde labeling of NG neurons, can provide answers to these questions. Moreover, our analysis demonstrates that not only does CCK signal predominantly through the right NG, as demonstrated by de Araujo and coworkers [6], but that GLP-1 and PYY do as well, and to the same extent. In contrast, neurotensin, GPR65, and 5-HT apparently signal equally through both sides of the NG.

Although several minor populations of neurons with various receptor expression patterns were detected—consistent with recent single-cell studies [14,24,30]—our RNAscope analysis identified the following two major, distinct populations of NG neurons co-expressing GPCR sensors for intestinal hormones: (1) a large population co-expressing receptors for GLP-1, CCK, and PYY (*Glp1r*, *Cckar*, and *Npy2r*) and (2) another large population expressing the neurotensin receptor, *Ntsr1*, along with *Gpr65* and often *5ht3a*.

### 3.1. Glp1r/Cckar/Npy2r Co-Expressing NG Neurons

The majority of *Glp1r*-positive neurons also express *Cckar* and *Npy2r*, while less than half of *Npy2r*-positive neurons also express *Glp1r* and *Cckar* (Figure 3). This discrepancy probably arises because a significant fraction of *Npy2r* neurons innervate the lungs rather than the GI tract [22]. Our re-analysis of the RNAseq data of Kupari and coworkers, with the correct annotation of *Glp1r*, demonstrated that the *Glp1r/Cckar/Npy2r* co-expressing neurons correspond to the NG13 and NG15 clusters in their atlas of NG neurons. These clusters also express *Trpv1* and are believed to innervate IGLEs in the gastro-intestinal muscle layers [24].

*Glp1r*-expressing NG neurons have convincingly been shown to respond to stretch, such as gastric dilatation, but, surprisingly, do not respond to food [13,14]. Our analysis also found *Glp1r* to be highly expressed in the NG8 and NG10 neurons, which are indicated as ‘polymodal mechano-sensors’ as they express the mechanosensory Piezo2, but no other gut hormone receptors [24]. Thus, although there are other minor populations of NG neurons expressing receptors for CCK, GLP-1, and PYY, these three gut hormone receptors are generally co-expressed in a major population of afferent vagal neurons, which might be subclassified depending on their expression of other genes. This finding suggests that CCK, mainly produced in the duodenum and upper small intestine, along with GLP-1 and PYY, which are both mainly produced in the distal small intestine and colon, could all be sensed by ‘a single type’ of afferent vagal neuron.

### 3.2. Individual Glp1r/Cckar/Npy2r Co-Expressing Neurons Can Apparently Sense the Entire GI Tract

Although the majority of tracer-labeled neurons were labeled by a single tracer, indicating innervation from either the stomach or colon, around 22.9% were labeled with both tracers, indicating that these neurons receive dendritic input from both regions (Figure 5). By combining retrograde labeling with RNAscope technology, we found that the double-tracer-labeled neurons expressed *Glp1r*, *Cckar*, and *Npy2r*, suggesting that they were GLP1R/CCKAR/NPY2R co-expressing neurons. In the stomach, *Glp1r*-expressing neurons densely innervate IGLEs and function as mechano-sensors [13,14]. Notably, the mucosa of the small intestine is innervated by both *Gpr65*- and *Glp1r*-expressing neurons. Except for the duodenum, the density of *Glp1r* neuronal innervation of the intestinal mucosa is comparable to that of *Gpr65* mucosal innervation (Figure 4 in [14]). GLP1R-expressing NG neurons also innervate the mucosa of the colon and rectum, although this innervation is sparse [14], aligning with the relatively low number of dually labeled neurons found in our retrograde tracing analysis involving the colon. Knight and coworkers, in their extensive analysis of vagal sensory neurons, found that, on average, 11% of labeled NG neurons innervated two segments of the murine GI tract, varying between 4% and 17% depending on the injection site [14]. However, they dismissed this to instead conclude that different regions of the GI tract are primarily innervated by distinct vagal sensory neurons [14]. While this may be true, approximately 10% of neurons follow the ‘one neuron to many regions’ innervation pattern, which, according to our dual retrograde labeling, covers the entire GI tract, with the neuronal co-expression of multiple gut hormone receptors and mechano-sensors, providing insights into the key questions mentioned above.

Thus, individual *Glp1r/Cckar/Npy2r* co-expressing NG neurons, which also express receptors like *Trpv1* (i.e., NG clusters 13 and 15), may receive mechano-sensory input from endings forming IGLEs in the muscle layer of the stomach. They may also receive chemosensory inputs from other dendrites innervating the mucosa of the upper small intestine, where their CCKA receptors sense CCK released from I cells in response to food. Additionally, the same neuron may receive chemosensory inputs from dendrites innervating the mucosa of the lower small intestine or even the colon, sensing GLP-1 and PYY secreted from L-cells through their GLP-1R and NPY2R receptors when a food bolus reaches those regions (Figure 6). It is unclear whether the indicated gut hormone receptors and mechano-receptors, or their mRNA, are transported into all the dendrites or only into the relevant ones. Importantly, illustrations of the GLP1R+ neurons innervating the gastric muscle layers are based on genetic reporters and do not display the actual receptor or its mRNA [13,14]. In the one-neuron-to-multiple-regions scenario, various mechanosensory and chemosensory signals from different parts of the GI tract are received and likely integrated already within these *Glp1r/Cckar/Npy2r* co-expressing NG neurons (Figure 6). Importantly, under physiological circumstances, all these signals, including stretch, are related to the presence of food in the GI tract.

The observation that individual vagal neurons co-expressing *Glp1r/Cckar/Npy2r* integrate signals from the entire GI tract may have significant clinical application. Specifically, it could enable the refinement of existing vagal nerve stimulation (VNS) therapies to selectively target these multimodal gut-sensing neurons, thereby improving the regulation of feeding behavior, mood, and metabolism.

### 3.3. Why Do GLP1R Neurons in the Literature Appear to Respond to Stretch Rather than Food?

Advanced in vivo neuronal imaging and functional studies have identified GLP-1R neurons as sensors of distention rather than food [13,14]. It is important to note that vagal afferent innervation by both ‘GLP1-R’ and ‘GPR65’ neurons is significantly denser in the stomach and the duodenum than in the lower small intestine and colon [14]. This suggests that, in the way recordings are made, it may be easier to obtain signals from neurons innervating the stomach and upper small intestine, potentially biasing the conclusions. There is no doubt that GLP1R neurons strongly innervate the stomach and respond to distention. However, according to our hypothesis, GLP-1R is likely a ‘marker’ of these multifunctional neurons that, in the stomach, primarily function as sensors of distention or stretch through their IGLE nerve endings in the muscle layer. We propose that GLP-1 receptors in nerve endings, dendrites of these multifunctional neurons which innervate the mucosa of the distal part of the small intestine, are activated by GLP-1 secreted in response to dietary nutrients when food reaches these locations. Future functional experiments are required to prove this hypothesis and to understand the physiological importance of these multifunctional neurons expressing multiple receptors and, for example, stretch sensors.

### 3.4. The GPR65/NTSR1 Co-Expressing NG Neurons and Their Role in Sensing Gut Hormones

In this study, we found that the vast majority of *Gpr65*-expressing neurons also express *Ntsr1*, and vice versa—almost all *Ntsr1*-expressing neurons also express *Gpr65* and are almost entirely distinct from *Glp1r*-expressing neurons (Figure 3b). Thus, the gut hormone neurotensin is sensed by what have been termed GPR65 neurons, constituting approximately 20% of all NG neurons (Figure 2). These neurons are a major population of afferent vagal neurons innervating the villi of the small intestine [13]. Our RNAscope analysis also demonstrates that a large fraction of these NTSR1/GPR65 neurons express 5ht3a (Figure 3c), consistent with functional single-neuron in vivo imaging studies [13]. These studies have demonstrated that GPR65/NTSR1 NG neurons sense food in the proximal small intestine, particularly in the duodenal bulb. The activation of GPR65 neurons was originally assumed to be mediated by 5-HT released from enterochromaffin (EC) cells [13]. However, unlike GLP-1, PYY, and neurotensin cells, EC cells of the small intestine do not express nutrient-sensing metabolite receptors such as FFAR1and GPR119 [31]. Importantly, EC cells express GLP-1 receptors, suggesting that 5-HT release from EC cells is likely stimulated by food through an indirect, paracrine mechanism mediated by GLP-1 from neighboring L-cells [31].

Neurotensin-secreting cells are located in the upper part of the intestinal villi, but originate in the crypts as initially GLP-1-producing L cells, which gradually change their hormone repertoire as they move up the crypt–villus axis, where they start to first synthesize and store PYY and eventually neurotensin [23]. Neurotensin is secreted in parallel with GLP-1 and PYY in response to dietary nutrient metabolites such as LCFAs, monoacyl-glycerol, amino acids, and glucose [23]. Based on our results, we argue that neurotensin directly conveys a signal about the presence of food in the lumen of the upper small intestine to the afferent vagal NTSR1/GPR65 neurons. In contrast, GLP-1 may stimulate these neurons indirectly by promoting 5-HT release from neighboring EC cells [31].

Although GPR65 is considered to be an orphan receptor, it is, in fact, an efficient pH, or rather a proton sensor [32]. Since GPR65/NTSR1 neurons particularly densely innervate the mucosa of the duodenal bulb [13], which regularly fills with highly acidic stomach contents during a meal, it can be envisioned that GPR65 senses these protons. However, it is generally not assumed that vagal neuronal endings reach the gut lumen. The secretion of bicarbonate by the epithelial cells in the duodenum to neutralize the acidic luminal content is associated with acidification on the basolateral side of the epithelium [33,34], which can be sensed by the dense network of vagal afferents expressing GPR65 in this region of the GI tract. However, GPR65/NTSR1 neurons are not restricted to the upper small intestine. Our retrograde labeling studies show that these neurons innervate the entire intestine, including the colon (Figure 4c). A low interstitial pH, which can potentially be sensed by GPR65, is a hallmark of intestinal inflammation [32]. This is particularly interesting because, according to Kupari and coworkers, who annotated these neurons as NG17 and NG18, NTSR1/GPR65 neurons express receptors for specific immuno/inflammatory mediators, such as cysteinyl leukotrienes (Cysltr2) and, in the NG18 cluster, the sphingosine 1-phosphate sensor S1pr3 [24]. Interestingly, neurotensin has also been associated with inflammatory conditions [35], although this is not yet well studied.

Thus, we propose that NTSR1/GPR65 neurons in the duodenum function as sensors of food [13,14] because their NTSR1 and 5HT3A receptors are indirectly stimulated by dietary nutrients, and their GPR65 receptors sense the food-associated gastric acid. However, in the rest of the intestine, NTSR1/GPR65 neurons likely monitor immune/inflammatory signals through the GPR65 sensing of interstitial acidity and, more specifically, through the sensing of immune/inflammatory mediators by Cysltr2 and S1pr3.

### 3.5. Gut Hormone Signaling Through Right Versus Left NG

While afferent vagal signaling has been proposed for nearly all gut hormones, CCK stands out as the classical example of neural gut–brain signaling [10,36]. Recently, de Araujo and coworkers demonstrated that CCK is the major mediator in a novel gut-to-brain reward neuronal pathway, leading to dopamine release in the striatum via the brainstem, and this pathway operates mainly, if not exclusively, through the right NG [6]. Consistent with this, we found that the right NG contains 50% more neurons expressing *Cckar* than the left. However, this pattern was also observed for *Glp1r* and *Npy2r*, but not for *Ntsr1*, *Gpr65*, and *5ht3a*. Although some *Cckar*-expressing neurons do not express *Glp1r* and *Npy2r*, it is tempting to suggest that *Cckar*–*Glp1r*–*Npy2r* co-expressing neurons are involved in de Araujo’s reward pathway, given the similar lateralization and preferential expression in the right NG for all three gut hormones receptors. Thus, we conclude that gut–brain signaling via afferent vagal neurons occurs mainly through the right NG for GLP-1 and PYY, similar to CCK.

This study focuses on the expression of specific gut hormone receptors in the NGs; however, a broader analysis of the potential asymmetrical expression of other gut-sensing hormone receptors is required. Additionally, the asymmetry in gut–brain axis signaling via the right NG should be further investigated in humans. Moreover, it remains to be determined whether the “one-neuron-to-multiple-regions” model also applies to gut–brain signaling in humans. Nevertheless, our findings have the potential to inform and improve VNS therapies for conditions related to gut dysregulation and weight management.

## 4. Materials and Methods

### 4.1. Animals

The animal experiments were conducted following the University animal care committee’s regulations. Glp1r reporter mice were generated by crossing the following two strains: Glp1r-CRE (Ozgene, Bentley DC, Australia) and B6.Cg-Gt(ROSA)^26Sortm14(CAG-tdTomato)Hze^/J (Jackson lab, Bar Harbor, ME, USA). In the resulting Glp1r-Cre/tdTomato-flox transgenic mice, all cells that express (or have previously been expressing) Glp1r produce tdTomato protein controlled by a constitutively active promoter [37]. The mice (12–20 weeks old) were housed under standard climate-controlled housing conditions with a 12 h light cycle and free access to water and mice chow. Wild-type (WT) C57Bl/6j mice of both sexes (10–12 weeks old) were obtained from Janvier Laboratory (France). For the study, a total of 7 Glp1r-Cre/tdTomato-flox transgenic mice and 71 WT C57Bl/6j mice were used, including 39 mice for tracer injections.

### 4.2. Tissue Preparation

For in situ hybridization and IHC, left and right nodose ganglia were dissected, fixed separately in 4% formaldehyde for 48 h, dehydrated in 25% sucrose, and frozen in O.C.T. compound (Sakura Finetek, Torrance, CA, USA). The frozen NG samples were cut into 8 μm thick sections in a way to make five series, which contained every fifth section of the nodose ganglion, and were mounted on Super Frost glass slides (Thermo Fisher Scientific, Waltham, MA, USA). For stereological quantification of the total number of neurons, the left and right ganglia were fixed separately for 48 h and embedded in paraffin. For qPCR, quickly dissected left and right ganglia were separately snap-frozen on dry ice and stored at −80 °C.

### 4.3. Retrograde Tracer Injection

For anatomical mapping, C57BL/6J male mice were anesthetized with isoflurane, treated with analgetic carprofen (5 mg/kg; s.c.), and a 1.5 cm long midline abdominal incision was made. In each mouse, a single segment of the GI tract (e.g., duodenum, jejunum, or proximal colon) was selectively labeled using 3–5 mice per region. The tracer (Alexa Fluor 647 conjugate of WGA; Invitrogen, Waltham, MA, USA) was dissolved in PBS and injected at multiple sites (up to 10) within the selected segment using a glass thin-wall capillary tube connected to a pico-injector (PLI-100A picoinjector, Multichannel Systems, Kusterdingen, Germany). A small total volume (~2 µL) of the solution was used for injection to minimize the spread of the tracer. For double-tracer experiments, two different tracers, Alexa Fluor 647 and Alexa Four 520 conjugate WGA (Invitrogen, Waltham, MA, USA), were injected in the colon and stomach, respectively. The microscopy-controlled injection was administered into the layer between the muscularis externa and the serosa layer. Then, the abdominal muscle and skin incisions were sutured, and mice were recovered. The animals were euthanized 48 h after the surgery, and their left and right nodose ganglia were dissected. The labeled neuronal terminals transported the tracer to their respective cell bodies, enabling the identification of nodose ganglion neurons that innervated the specific GI segment injected. On average, seven sections from each ganglion were analyzed. The number of tracer-positive neurons were normalized to the total number of HuC/HuD-positive neurons, labeled as described below.

### 4.4. Immunohistochemistry

To quantify GLP1R-positive neurons in the nodose ganglia, 4–11 cryosections per ganglion collected from seven mice were boiled in citrate buffer (pH 6) for 10 min, blocked in 5% donkey serum/0.1% Triton X100/PBS for 1 h, and then incubated overnight with primary rabbit anti-GLP1r antibody (1:1000; #ab218532 Abcam, Cambridge, UK) combined with mouse anti-HuC/HuD antibody (1:500; #A21271 ThermoFisher Scientific, Waltham, MA, USA) at 4 °C. The sections were washed with PBS, incubated with a mixture of corresponding secondary antibodies conjugated to Alexa Fluor 568 or Alexa Fluor 488 (1:800; #A10042 and #A21202 ThermoFisher Scientific), and coverslipped using DAPI-supplemented ProLong Gold Antifade medium (Life Technologies, Carlsbad, CA, USA). As a negative control, adjacent sections were processed simultaneously, but the primary antibody was ommited. Images were recorded using a confocal microscope (Zeizz NL700; Zeizz, Jena, Germany). The quantification of tdTomato+-cells and total Huc/Hud+-neurons was performed with ImageJ (version 1.53a).

To quantify the number of neurons with retrograde tracer, nodose ganglia were serially cut into sections, blocked in the aforementioned blocking buffer for 1 h, and then incubated overnight with primary mouse anti-HuC/HuD antibody (1:500; ThermoFisher Scientific). The sections were washed with PBS, incubated with a secondary antibody conjugated to Alexa Fluor 488 (1:800; ThermoFisher Scientific), and coverslipped using DAPI-supplemented ProLong Gold Antifade medium (Life Technologies). The percentage of retrograde tracer-positive neurons in the left and right nodose ganglia was quantified on images that were captured by Zeiss AxioObserver microscope with a 20× objective, as a ratio of tracer-positive neurons to the total number of neurons and presented in percent.

### 4.5. Stereology

A quantitative analysis of the total number of neurons in the NGs (n = 7, WT C57Bl/6j mice) was performed using the unbiased optical fractionator [38]. Every 30 μm thick serial section was deparaffinized and stained with 1% Cresyl Violet. The stereological counting of neurons in the left and right nodose ganglion was performed with CAST-GRID software, version 5.6 (Olympus, Søborg, Denmark) using an Olympus BX50 microscope equipped with a 100× oil-immersion objective and a camera. In each section, the reference space was outlined at a low magnification (10×) to indicate the area for dissector sampling. A series of systematic random sampling sites with a 100 µm distance between them was used to estimate the total number of Cresyl-Violet-stained neurons at a higher magnification (100×). The section thickness was measured at every fifth sampling site. An average of 90 dissector probes per ganglion were sampled.

### 4.6. RNAscope

For the detection of selected targets, commercially available kits, including RNAscope chromogenic “red” assay, RNAscope chromogenic Duplex assay, and RNAscope multiplex fluorescent assay (all from Advanced Cell Diagnostics, Hayward, CA, USA), were used according to the manufacturer’s protocols [39]. Deparaffinized sections were washed in PBS, treated with a peroxidase blocker, and boiled for 2 min in a pretreatment solution. After digestion with Protease Plus solution for 30 min at 40 °C, target probes, including positive and negative control probes (Table A1; Advanced Cell Diagnostics, USA), were hybridized for 2 h at 40 °C in the HybEZ Hybridization System (Advanced Cell Diagnostics), followed by a series of signal amplification and washing steps. Bound probes were detected by either chromogenic reaction using red and/or green chromogens or by fluorescent labeling using Opal^TM^ 520, Opal^TM^ 570, and Opal^TM^ 690 reagents (Perkin Elmer, Waltham, MA, USA). Images were recorded on Zeiss LSM700 confocal microscope and processed with Visiopharm software (newCAST v. 3.4.1.0, Visiopharm, Horsholm, Denmark) or ImageJ software [40].

### 4.7. Gene Expression Analysis

The total RNA was isolated from left and right nodose ganglia (n = 5) using a Qiagen RNeasy micro kit (Qiagen, Hilden, Germany). cDNA was synthesized using a SuperScript III Reverse Transcription Kit (Invitrogen) according to the manufacturer’s instructions. Real-time qPCR was performed with 4 ng cDNA per 10 µL reaction volume using the Fast SYBR Green Master Mix (Applied Biosystems, Waltham, MA, USA) on a LightCycler480 (Roche, Basel, Switzerland). Samples were analyzed in duplicates and the reaction started with pre-incubation at 95 °C for 2 min. This was continued with 45 cycles as follows: 95 °C for 15 s and 60 °C for 45 s. The reaction then proceeded with a final following cycle: 95 °C for 20 s, 60 °C for 1 min, and 95 °C for 15 s, concluding with the finale stage at 40 °C for 30 s. The relative gene expression was calculated using 2^−(CTgene−CTreference)^, where CT reference is the mean value of three reference genes, *Ywhaz*, *β-actin*, and *GAPDH*. The sequences of primers are listed in Table A2.

### 4.8. Glp1r Single-Cell Data Analysis

Using publicly available nodose ganglia scRNA-seq raw data [24], we converted the genomic coordinates to the genome build mm10 using the liftOver tool (https://genome-store.ucsc.edu/, accessed on 30 September 2019) [41] and applied the PolyA_DB database version 3.2 [42] to identify the genomic coordinates for alternative polyadenylation sites within the *Glp1r* gene. We re-annotated the transcription end coordinate for *Glp1r* to include the most downstream polyadenylation site and re-computed the counts for *Glp1r* using Cell Ranger version 3.0.0 with the genome build mm10.

### 4.9. Nodose Ganglia Explants Preparation and Field Nerve Potential Recording

To perform ex vivo field nerve potential recordings, the left and right ganglia with ~0.5 cm vagus nerve fibers were dissected from WT mice (n = 6) of both sexes, placed on semi-porous culture well inserts, and maintained for 1–2 days in Neurobasal-A medium supplemented with 2% B-27, 2 mM GlutaMAX (all from ThermoFisher Scientific), 0.5 μM T3, 0.5 μM T4 (both from Sigma, St. Louis, MO, USA), 200 U/mL penicillin, 5 μg/mL streptomycin, and 10 mM HEPES, as described previously [29]. The CCK-8 (10–20 nM; TOCRIS, Bristol, UK), GLP-1 (10–100 nM; TOCRIS), and Exendin-4 (10–100 nM; TOCRIS) peptides were gravity flow-applied for 10 min in aCSF, whereas KCl (55 mM; Sigma Aldrich, St. Louis, MO, USA) was gravity applied for 3 min. In vitro field potentials were recorded with a custom-built nerve amplifier (50,000×), filtered at DC-2 KHz, and digitized (5 KHz) by a PCI-6289, M Series A/D-board (National Instruments, Austin, TX, USA) controlled by Igor Pro 8.04 (Wavemetrics, Lake Oswego, OR, USA). The raw nerve signals were high-pass filtered off-line (Hanning Finite Impulse Response digital filter, start of pass band: 1000 Hz), rectified, and smoothed (Savitzky–Golay smoothing algorithm with a smoothing order of 2 over 501 points).

For Ca^2+^ imaging, the isolated nodose ganglia (n = 3, WT mice) were transduced with the AAV construct pGP-AAV-syn-jGCaMP7s-WPRE (AAV9, Addgene, Watertown, MA, USA, Cat#:104487-AAV9), coded for a genetically encoded Ca^2+^ sensor (GECI) driven by the synapsin promoter, and used for experiments after one week in culture [43]. The NG explants were stimulated with CCK-8 (10 µM) and Ca^2+^ imaging was performed on a stereomicroscope (Leica MZ16; Leica, Germany) illuminated by an external light source for fluorescence excitation (Leica EL6000). For GECI, green channel fluorescence was visualized using a bandpass filter (GFP3: excitation: 470/40 nm, barrier filter 525/50 nm). Live image stacks were captured by an EMCCD camera (Andor Luca EM S DL-658M, Andor Technology, Belfast, UK), controlled by the SOLIS software (Version 4.24.30004.0, Andor Technology, Belfast, UK).

### 4.10. Statistical Analysis

Statistics were performed in GraphPad Prism version 9.4 sofware (GraphPad, San Diego, CA, USA), and data are expressed as mean values with SEM indicated. The statistical significance between left and right nodose ganglion was tested with a two-tailed, paired Student’s *t*-test and defined as *p* < 0.05. *, **, ***, and **** indicate *p* < 0.05, *p* < 0.01, *p* < 0.001, and *p* < 0.0001, respectively.

## Figures and Tables

**Figure 1 ijms-26-02053-f001:**
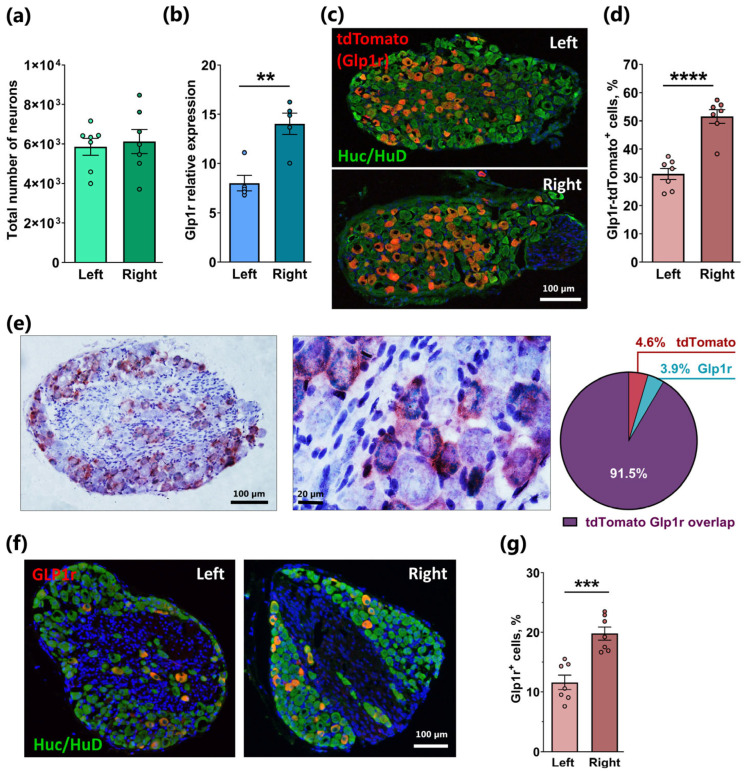
Glucagon-like peptide-1 receptor expression and distribution in the murine nodose ganglion. (**a**) Stereological quantification of Cresyl-Violet-stained neurons in the right and left mouse nodose ganglia (n = 7). (**b**) The expression level of Glp1 receptors in the right and left nodose ganglia of wild-type (WT) mice (n = 7). (**c**) Representative images from the left and right nodose ganglia of Glp1r-tdTom mouse, where Cre expression is reported by tdTomato (red) followed by counterstaining for HuC/HuD pan-neuronal marker (green). (**d**) Quantification of tdTomato+-neurons normalized to the total HuC/HuD+-neurons in the right and left nodose ganglia (n = 7). ((**e**), left) Representative image of NG at low and high resolution after Duplex chromogenic RNAscope for *tdTomato* (red) and *Glp1r* (blue). Bright-field images were recorded from the nodose ganglia of Glp1r-tdTom mouse. Sections were counterstained with hematoxylin. ((**e**), right). Pie charts showing the percentage of single-labeled positive cells (red or blue) and double-labeled positive cells (violet), indicating that the majority of *tdTomato*+-cells are also positive for *Glp1r* labeling. (**f**) Representative images from the left and right nodose ganglia co-immunostained for GLP1r and HuC/HuD. (**g**) Quantification of the GLP1r+-neurons in the right and left nodose ganglia normalized to the total Huc/HuD+-neurons (n = 7). In (**a**,**b**,**d**,**g**), data are presented as means ± SEM. **, ***, and **** indicate *p* < 0.01, *p* < 0.001, and *p* < 0.0001, respectively.

**Figure 2 ijms-26-02053-f002:**
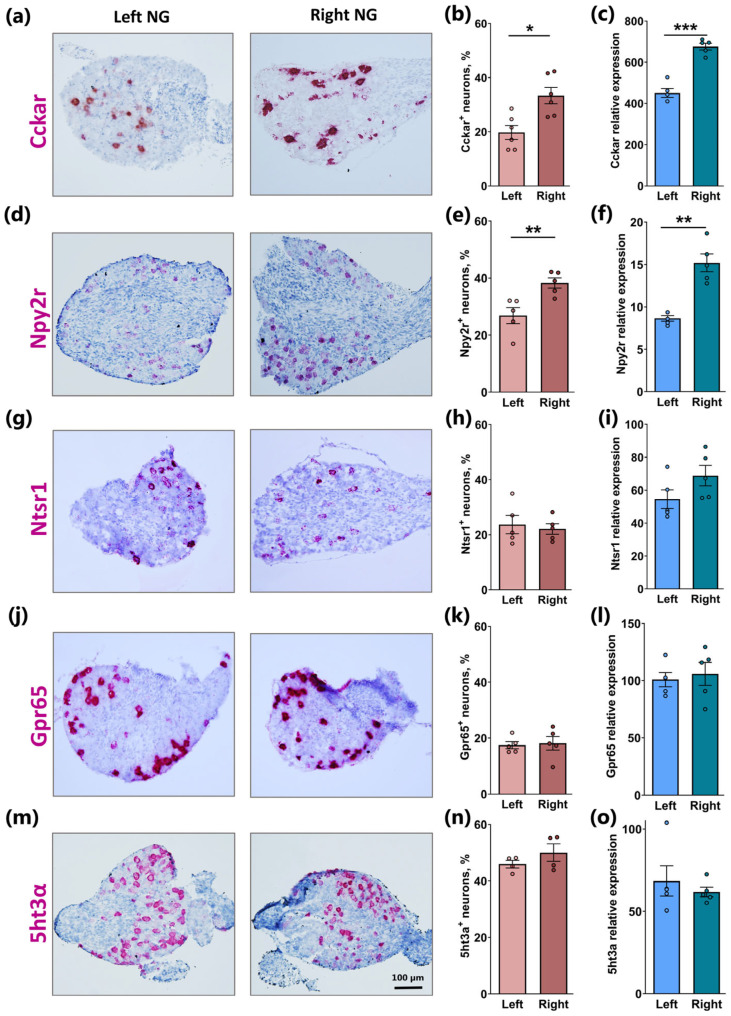
Gut hormone receptor expression in left and right nodose ganglia. (**a**,**d**,**g**,**j**,**m**) Representative images of chromogenic RNAscope for *Cckar*, *Npy2r*, *Ntsr1*, *Gpr65*, and *5ht3α* in the left and right nodose ganglia. Bright-field images were collected from the nodose ganglia of wild-type mice. Sections were counterstained with hematoxylin. (**b**,**e**,**h**,**k**,**n**) Quantification of the percentage of neurons in the right and left nodose ganglia positive for *Cckar*, *Npy2r*, *Ntsr1*, *Gpr65*, and *5ht3α* (n = 4–6). (**c**,**f**,**i**,**l**,**o**) Relative mRNA expressions of *Cckar*, *Npy2r*, *Ntsr1*, *Gpr65*, and *5ht3α* in the left and right nodose ganglia of WT mice measured using qPCR (n = 5). Data are presented as means ± SEM. *, **, and ***, indicate *p* < 0.05, *p* < 0.01, and *p* < 0.001, respectively.

**Figure 3 ijms-26-02053-f003:**
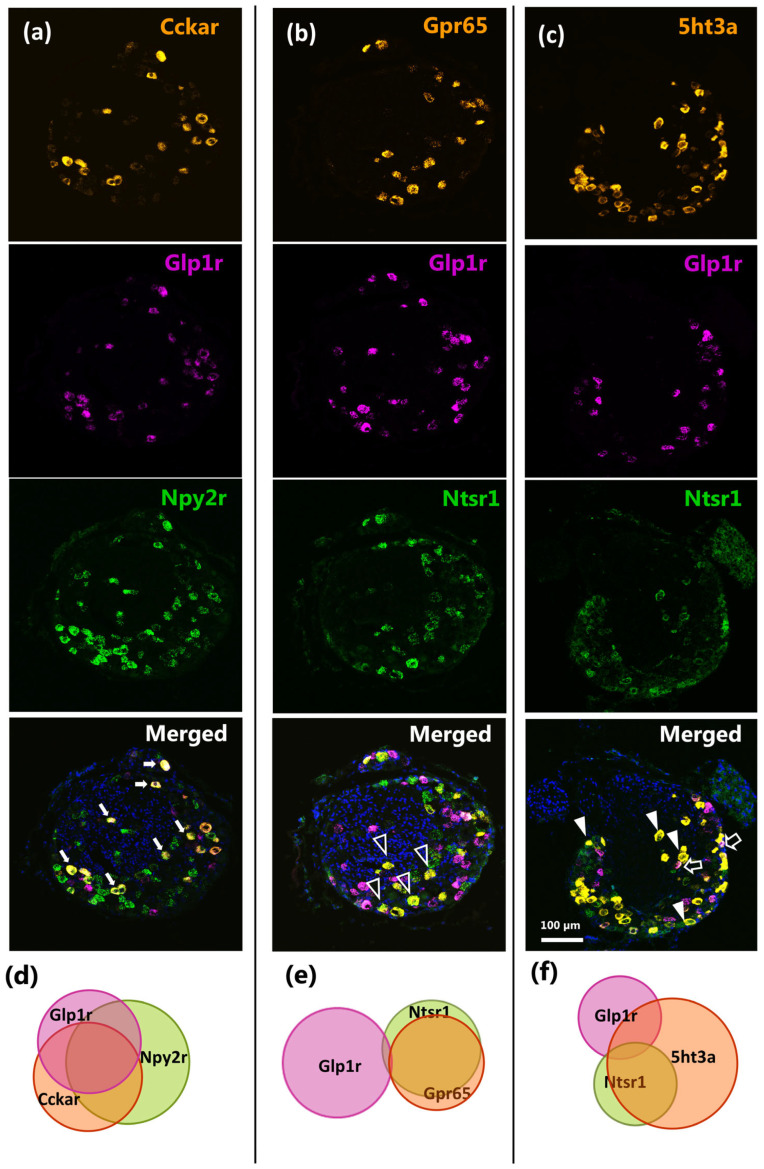
Triple fluorescent in situ hybridization for gut hormone receptors in the nodose ganglia. (**a**) Representative images of multiplex RNAscope for *Cckar*, *Glp1r*, and *Npy2r*. Arrows indicate representative triple-labeled cells. (**b**) Representative images of multiplex RNAscope for *Gpr65*, *Glp1r*, and *Ntsr1*. Empty triangles indicate examples of *Gpr65*/*Ntsr1* double-labeled cells (**c**) Representative images of multiplex RNAscope for *5htr3α*, *Glp1r*, and *Ntsr1*. Triangles and empty arrows indicate examples of *5ht3α/Ntsr1* and *Glp1r/5ht3α* double-labeled cells, respectively. (**d**,**e**,**f**) Semi-quantitative Venn diagrams depict the relative sizes of subpopulations and relation between gut hormone receptor expressions in murine nodose ganglia. The diameter of each circle represents the relative number of NG cells positive for the corresponding gene (n = 3).

**Figure 4 ijms-26-02053-f004:**
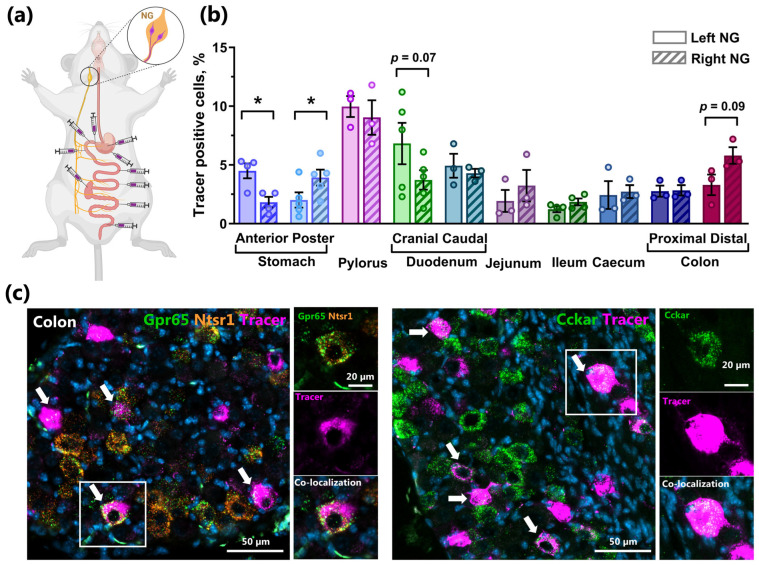
Region-dependent innervation of the gut by left and right nodose ganglia. (**a**) Schematic overview of retrograde tracer injections. (**b**) Quantification of the percentage of tracer-positive cells in the left and right nodose ganglia 48 h after the tracer was injected in the indicated gut region (n = 3–5 per region). Data are normalized to HuC/HuD+-neurons and presented as means ± SEM. * indicate *p* < 0.05. (**c**) Representative images of NG labeled with a retrograde tracer, Alexa Fluor 647 conjugated WGA, injected in colon, and stained with probes for *Ntsr* and *Gpr65* (**left panel**) and *Cckar* (**right panel**). Arrows indicate neurons co-labeled with both the tracer and probes.

**Figure 5 ijms-26-02053-f005:**
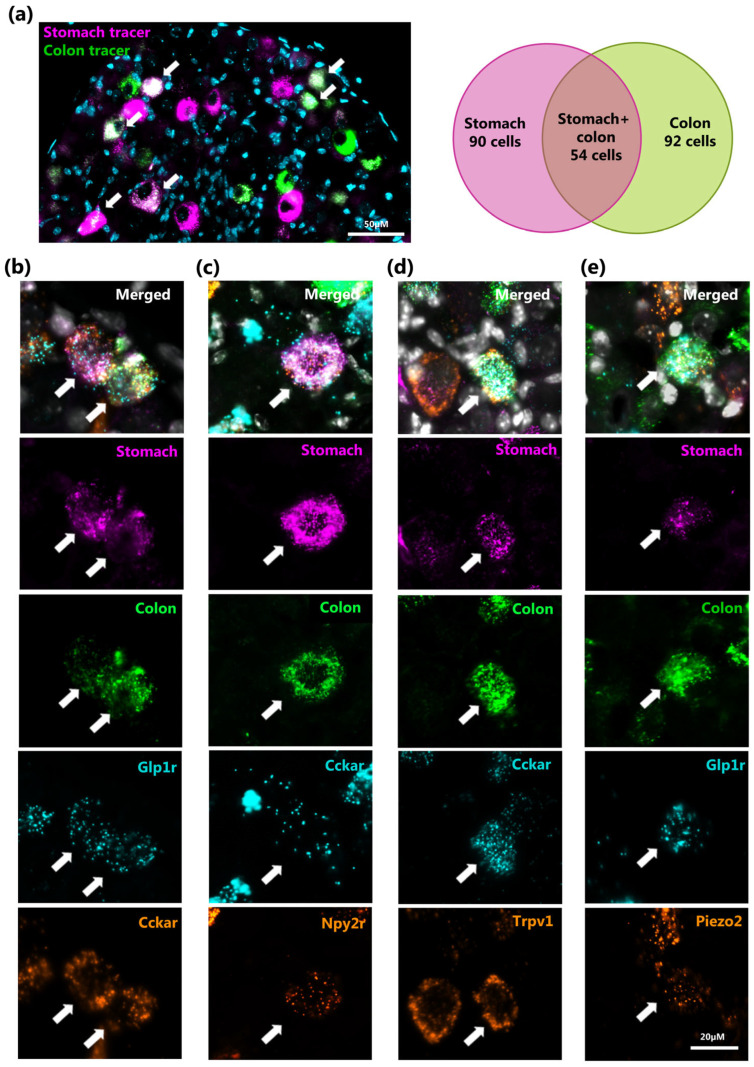
Co-expression of gut-hormones and mechanoreceptors in neurons that innervate both the stomach and colon simultaneously. (**a**) A section of the nodose ganglion shows the distribution of neurons retrogradely labeled from the stomach and colon (**right**). Double positive neurons were indicated with arrows. Venn diagrams depict the relative sizes of NG neuronal subpopulations retrogradely labeled by two different tracers injected in the stomach and colon (**left**). (**b**–**e**) Examples of NG neurons with dual retrograde labeling from the stomach and colon (magenta and green, respectively), which simultaneously express *Glp1r* (cyan) and *Cckar* (orange) (**b**); *Cckar* (cyan) and *Npy2r* (orange) (**c**); *Cckar* (cyan) and *Trpv1* (orange) (**d**); and *Glp1r* (cyan) and *Piezo2* (orange) (**e**). Neurons, positive for all four markerswhere indicated with arrows.

**Figure 6 ijms-26-02053-f006:**
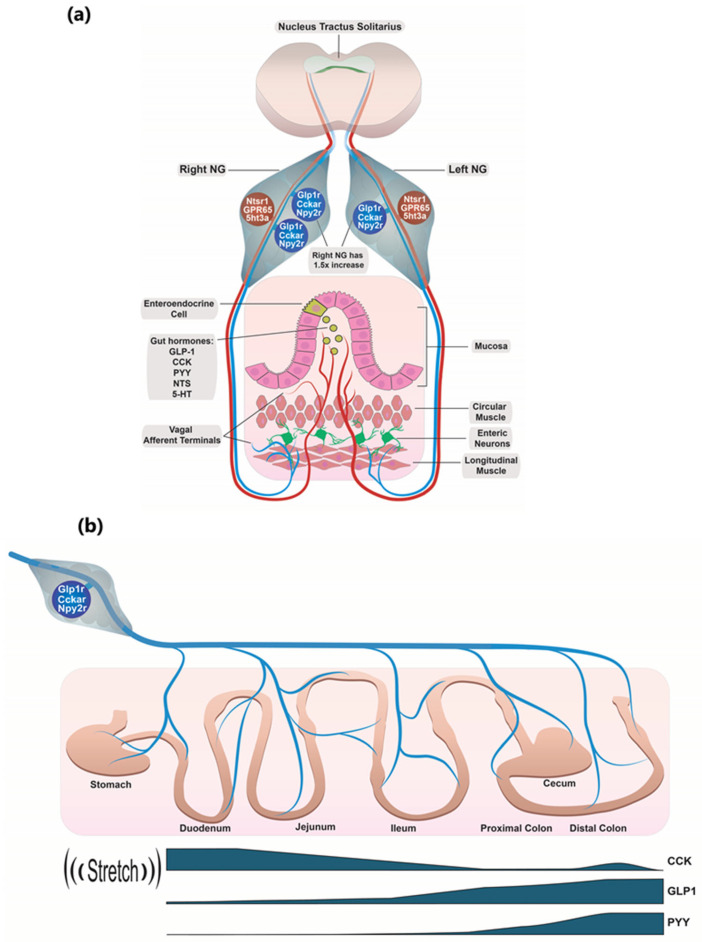
Schematic overview of expression of gut hormone receptors by afferent vagal neurons that innervate GI tract. (**a**) This study identified two main populations of vagal neurons. One of them carries *Ntsr1* and *Gpr65* and innervates mucosa of GI tract and the second is enriched with *Glp1r*, *Cckar*, and *Npy2r* and predominantly innervates muscularis, whereas both populations carry *5ht3α*. (**b**) Schematic mechanism of GIT innervation by nodose neurons. Cells that express gut hormone receptors can simultaneously innervate the stomach detecting stretch there and receive signals from EECs in various segments of the intestine.

## Data Availability

Data is contained within the article and Appendix A.

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
