# Peer review of "Neurons Co-Expressing GLP-1, CCK, and PYY Receptors Particularly in Right Nodose Ganglion and Innervating Entire GI Tract in Mice"

_ijms, 2025, doi:10.3390/ijms26052053_

Round 1

Reviewer 1 Report

Comments and Suggestions for Authors

The study proposes a novel approach to the lateralization of receptors for gastrointestinal hormones (CCK, GLP-1, PYY) in the right and left nodose ganglia. The objectives are well-defined and justify the choice of techniques such as RNAscope and retrograde tracing, but there are some aspects that should be address before publication:

Minor revisions

Reagents: missing some data and supplier

Venn diagrams: have more explanation in the legend

Increase the resolution of certain images to facilitate coexpression analysis in small details. e.g. Figure 4 a and b. There are some words cut off. Figure 2: the legend is moved.

Include more descriptive legends in the tables, especially those that present technical data such as primer sequences. Amplicon size, temperatures, efficiency percentage and if the sequence has been previously published.

Major revisions

-qPCR and gene expression analysis: 

cDNA synthesis: Total RNA was quantified? Quantity of RNA per reaction?

Missing data on amplicon size and primer efficiency. Melting and PCR cycling temperature. If the reactions were done in triplicate or technical duplicates. 

-For IHC and RNA scope, it is not clear to me if positive and negative controls of the technique were done. Clarify in the text

-Retrograde tracer injection: did they take into account the weight of the animals? Why is it said that approximately 2ul were injected?

-Was any program used for the statistical analysis? How were the Venn diagrams constructed?

Suggestions for further discussion: Although possible clinical applications are mentioned, the direct impact on the physiological functionality of the observed asymmetry could be developed further.

Author Response

Comments: The study proposes a novel approach to the lateralization of receptors for gastrointestinal hormones (CCK, GLP-1, PYY) in the right and left nodose ganglia. The objectives are well-defined and justify the choice of techniques such as RNAscope and retrograde tracing, but there are some aspects that should be address before publication:

Minor revisions

  • Reagents: missing some data and supplier

Author reply: We are thankful to the reviewer for highlighting these points. Additional information on the origin of reagents and details regarding the concentrations used have now been incorporated into the Material and Methods section.

  • Venn diagrams: have more explanation in the legend

Author reply: In response to the reviewer's comment, additional information has now been provided in the Legend for Figure 3, p. 9.

  • Increased the resolution of certain images to facilitate coexpression analysis in small details. e.g. Figure 4 a and b. There are some words cut off. Figure 2: the legend is moved.

Author reply: In line with reviewer' suggestion, the resolution of Figures 4 a and b has been improved, and the embedded text has been corrected. Additionally, the Legend for Figure 2 has been revised accordingly.  

  • Include more descriptive legends in the tables, especially those that present technical data such as primer sequences. Amplicon size, temperatures, efficiency percentage and if the sequence has been previously published.

Author reply: In response to reviewer feedback, we have provided detailed information on the amplicon size, which is now indicated in Table A2. The primers for qPCR analysis were designed in-house using Primer3 software [Untergasser A, Cutcutache I, Koressaar T, Ye J, Faircloth BC, Remm M and Rozen SG. Primer3-new capabilities and interfaces. Nucleic Acids Res. 2012 Aug 1;40(15):e115]. Primer’ specificity was confirmed using Primer-Blast (NCBI.nlm.nih.gov) with NCBI Messenger RNA Reference Sequences (Mus musculus). Only primer sets with an efficiency between 90%-110% were used for the final analysis. This information has been included in the legend for Table A2. Since the melting temperature (Tm) for each primer varies and can be easily determined using various primer design software, we have not included this information in the table. The primers for the three reference genes have been validated in our lab and are routinely used in qPCR. However, due to journal policies limiting self-references to a maximum of eight—which we have already reached in the main text—we have opted not to include references for the primers in the table.

In addition, we have extended Table A1 to include information on the temperature and duration of in situ hybridization.  

Major revisions

  • qPCR and gene expression analysis: cDNA synthesis: Total RNA was quantified? Quantity of RNA per reaction?

Missing data on amplicon size and primer efficiency. Melting and PCR cycling temperature. If the reactions were done in triplicate or technical duplicates. 

Author reply: We appreciate the reviewer for highlighting this important point. The qPCR reaction parameters, along with the cDNA amount per reaction, are now included in the Material and Methods, p. 18. Samples were analysed in duplicates, and this information has now been added accordingly. As mentioned earlier, the amplicon size for each primer set is now provided in Table A2. Additionally, details regarding primer efficiency have been included in the legend for Table A2.

  • For IHC and RNA scope, it is not clear to me if positive and negative controls of the technique were done. Clarify in the text.

Author reply: We are grateful to the reviewer for highlighting this important methodological point. We have now expanded the Material and Methods section to include details on the positive and negative controls for RNAscope analysis (p. 18) and have added the corresponding information in Table A1. For the IHC protocol, the omission of the primary antibody served as a negative control. The text has been modified accordingly, p. 17.

  • Retrograde tracer injection: did they take into account the weight of the animals? Why is it said that approximately 2ul were injected?

Author reply: The body weight of mice was within a very narrow range (21-23 g) and, therefore, was not considered as a variable in our study. For microinjections, approximately 2 µL of total volume was delivered across multiple sites within the gut (about 10 sites per animal’ gut region). Although 2 µL was initially loaded into the microinjector, minor microleakages were unavoidable due to the setup of the procedure. As a result, we used the term 'approximately' to account for slight variations in the exact injected volume.

  • Was any program used for the statistical analysis? How were the Venn diagrams constructed?

Author reply: For statistical analysis, we have used GraphPad Prism, which is now indicated in the Material and Methods,  p. 19. Venn diagrams were created manually, with the diameter of each circle representing the relative number of NG neurons positive for the corresponding gene. This information has been added to the Legend of Figure 3, p. 9. Additionally, we have modified the Venn diagram in Figure 3d to enhance the clarity of the presented relationships.

Suggestions for further discussion: Although possible clinical applications are mentioned, the direct impact on the physiological functionality of the observed asymmetry could be developed further.

 Author reply: We sincerely appreciate the reviewer's thoughtful suggestion. While our findings indicate a degree of lateralization in vagal signaling pathways, drawing definitive conclusions about the functional implications of this asymmetry would be highly speculative at this stage. Our study was not specifically designed to assess functional differences between the left and right nodose ganglia in physiological regulation. Therefore, extrapolating from our data without dedicated functional experiments would extend beyond the intended scope of our current analysis. Additionally, our discussion already provides a detailed interpretation of our findings in the context of previous studies, addressing key mechanistic questions related to gut hormone signaling. Expanding the discussion further to hypothesize potential functional consequences of asymmetry would substantially lengthen an already comprehensive and complex text without providing definitive new insights. From our perspective, such an extension is not essential to the core objectives of this study, which primarily focus on mapping receptor expression patterns and identifying major populations of vagal afferent neurons. A thorough investigation of the functional implications of left-right asymmetry would require targeted experimental approaches, which we consider an important direction for future research. For these reasons, we have chosen to limit our discussion to conclusions directly supported by our data, avoiding speculation on functional consequences that remain to be experimentally explored.

Reviewer 2 Report

Comments and Suggestions for Authors

The manuscript submitted for review is an interesting scientific and research work on the localization and co-expression of selected receptors for gastrointestinal hormones in the membrane of neurons located in the nodose ganglia. The work contains a lot of data on both the distribution and immunohistochemical characteristics of neurons located in the nodose ganglia regarding the distribution and colocalization of receptors in the membrane of these neurons for hormones such as GLP-1, CCK PYY, serotonin or neurotensin. The work uses numerous methods to confirm both quantitatively and qualitatively the presence and distribution of the studied receptors. However, I have several questions and comments regarding both the presented research results, especially regarding neuronal labelling, and the description of the materials and methods, which I include below:

Abstract

Line 22 – explain in full name an abbreviation GLP1R

Line 26, 27 - explain in full name abbreviations CckaR, Npy2r, Ntsr, Gpr65, 5ht3a itd

Please explain all abbreviations appearing for the first time in the manuscript text with their full names.

Introduction

Line 57, 59 - explain in full name abbreviations 5-HT, CCK, GLP-1, PYY,

Line 69 - explain in full name an abbreviation GPR65

Line 96 - explain in full name an abbreviation GPCR

From line 104 “Our results indicate that not only CCK but also GLP-1 and PYY primarily signal through the right NG, whereas receptors for neurotensin, GPR65 and 5-HT were equally distributed among the two ganglia. Furthermore, retrograde tracing demonstrated that a relatively large population of afferent vagal neurons co-expressing the receptors for CCK, GLP-1 and PYY innervates the entire GI-tract from the stomach to the colon. These findings could potentially explain e.g. the enigmatic presence of GLP-1 receptor-expressing neurons in the muscle layer of stomach, part of the GIT 110 where GLP-1 is not produced”  - in my opinion this part of the introduction should be a summary and should be posted as a conclusion

Results

2.1. Asymmetrical expression of Glp1r in the left and right nodose ganglion

no information on how many animals were used to conduct this study

Line 121 - explain in full name an abbreviation IHC

Line 175 - explain in full name an abbreviation NPY

Figure 2 - explain with full name all abbreviations that appear for the first time in the description

Line 300 – should be 5Htr3a

Line 301 - explain in full name an abbreviation Trpv1

2.5. Retrograde tracing demonstrates NG lateralization of stomach innervation and strong innervation of the distal intestine in mice

- How did you recognize which population of labelled neurons in the NG is responsible for innervating a specific part of the gastrointestinal tract?

If one tracer was used for labelling, how was it recognized which population of labelled neurons in the NG ganglion is responsible for innervating, for example, the stomach, and which the duodenum or other parts of the intestine?

Was the tracer administered to specific sections of the gastrointestinal tract in different animals to isolate the population of neurons in the NG supplying a specific part of it?

Unfortunately, this information is not included in the part concerning the description of the tracing methodology.

Line 328 - explain in full name an abbreviation EECs

Materials and Methods

4.1. Animals

- how many mice of each type were used for the planned experiments

4.2. Tissue preparation

- what number of ganglia (how many animals were used) in immunohistochemical and in situ hybridization studies as well as in stereological quantification and for qPCR

4.3. Retrograde tracer injection

How many mice were used for the nodal ganglion neuron labelling procedure?

4.4. Immunohistochemistry

- how many sections were stained to determine the number of GLP1R neurons?

- how many and how were the sections counted for the presence of traced neurons, how many animals were used to perform this study?

4.5. Stereology

How many mice were used for this study?

4.9. Nodose ganglia explants preparation and field nerve potential recording

How many mice were used for this study?

References

All literature items should be prepared in accordance with the rules required by the journal.

Line 132 - after the literature item number 18 appears in the text number 27 instead of number 19 in the order

no citation of references 19 to 26 in the manuscript text

there is no order of numbering of cited literature items in the manuscript text, this should be sorted out as a whole

Author Response

The manuscript submitted for review is an interesting scientific and research work on the localization and co-expression of selected receptors for gastrointestinal hormones in the membrane of neurons located in the nodose ganglia. The work contains a lot of data on both the distribution and immunohistochemical characteristics of neurons located in the nodose ganglia regarding the distribution and colocalization of receptors in the membrane of these neurons for hormones such as GLP-1, CCK PYY, serotonin or neurotensin. The work uses numerous methods to confirm both quantitatively and qualitatively the presence and distribution of the studied receptors. However, I have several questions and comments regarding both the presented research results, especially regarding neuronal labelling, and the description of the materials and methods, which I include below:

Abstract

Line 22 – explain in full name an abbreviation GLP1R

Line 26, 27 - explain in full name abbreviations CckaR, Npy2r, Ntsr, Gpr65, 5ht3a itd

Please explain all abbreviations appearing for the first time in the manuscript text with their full names.

Introduction

Line 57, 59 - explain in full name abbreviations 5-HT, CCK, GLP-1, PYY,

Line 69 - explain in full name an abbreviation GPR65

Line 96 - explain in full name an abbreviation GPCR

Author reply: We appreciate the reviewer for bringing these issues to our attention. Following reviewer’s note, the necessary corrections have been made throughout the text of entire manuscript, including the Abstract.

From line 104 “Our results indicate that not only CCK but also GLP-1 and PYY primarily signal through the right NG, whereas receptors for neurotensin, GPR65 and 5-HT were equally distributed among the two ganglia. Furthermore, retrograde tracing demonstrated that a relatively large population of afferent vagal neurons co-expressing the receptors for CCK, GLP-1 and PYY innervates the entire GI-tract from the stomach to the colon. These findings could potentially explain e.g. the enigmatic presence of GLP-1 receptor-expressing neurons in the muscle layer of stomach, part of the GIT 110 where GLP-1 is not produced”  - in my opinion this part of the introduction should be a summary and should be posted as a conclusion

Author reply: Indeed, this text summarizes our findings and, in principle, could be part of the Conclusion section. However, according to the journal's guidelines, the Conclusion section is the final part of the manuscript, which is following the Materials and Methods, i.e. section 5. We believe that placing this information there may reduce its visibility and impact, as it could be overlooked by readers. Therefore, we have positioned it at the end of the Introduction to ensure it receives appropriate attention. We hope the reviewer understands and appreciates our reasoning.

Results

2.1. Asymmetrical expression of Glp1r in the left and right nodose ganglion

no information on how many animals were used to conduct this study

Line 121 - explain in full name an abbreviation IHC

Line 175 - explain in full name an abbreviation NPY

Figure 2 - explain with full name all abbreviations that appear for the first time in the description

Line 300 – should be 5Htr3a

Line 301 - explain in full name an abbreviation Trpv1

Author reply: In response to reviewer’s comment, we have clarified that 'n = 7,' as mentioned throughout Section 2.1 and in the Legend of Figure 1, refers to the number of mice, see p. 3, line 122. The full name for IHC has now been provided, p. 3, line 126. Additionally, as noted earlier, full names for abbreviations have been included in the text: neuropeptide Y (NPY) on p. 5, line 180, while abbreviations mentioned in the Legend of Figure 2 were first introduced in the Abstract. The full name of WT has also been added now to the Legend of Figure 1, p. 4, line 146.

2.5. Retrograde tracing demonstrates NG lateralization of stomach innervation and strong innervation of the distal intestine in mice

- How did you recognize which population of labelled neurons in the NG is responsible for innervating a specific part of the gastrointestinal tract?

If one tracer was used for labelling, how was it recognized which population of labelled neurons in the NG ganglion is responsible for innervating, for example, the stomach, and which the duodenum or other parts of the intestine?

Was the tracer administered to specific sections of the gastrointestinal tract in different animals to isolate the population of neurons in the NG supplying a specific part of it?

Unfortunately, this information is not included in the part concerning the description of the tracing methodology.

Author reply: Indeed, for anatomical mapping, the tracer was injected into specific regions of GIT, using 3-5 mice per region. We have now extended the methodological section to provide additional clarification, see pages 16-17. We have also expanded on the proposed mechanism of nodose neuron labeling following tracer injection. This information is now included in the Material and Methods, p. 17, lines 568-570.

Line 328 - explain in full name an abbreviation EECs

Author reply: Full name for the abbreviation EECs is now provided on p. 9, line 313.

Materials and Methods

4.1. Animals

- how many mice of each type were used for the planned experiments

Author reply: In total, seven Glp1r-Cre/tdTomato-flox transgenic mice and 71 WT C57Bl/6j mice were used. This information is now included in the Material and Methods, p. 16, lines 542-544.

4.2. Tissue preparation

- what number of ganglia (how many animals were used) in immunohistochemical and in situ hybridization studies as well as in stereological quantification and for qPCR

4.3. Retrograde tracer injection

How many mice were used for the nodal ganglion neuron labelling procedure?

Author reply: For Glp1r IHC, 4-11 cryosections per ganglion were collected from seven mice, as now indicated in the Material and Methods, p. 17, line 575.

For stereological quantification, nodose ganglia from seven WT C57Bl/6j mice were used, as mentioned in the Material and Methods, p. 17, line 597.

For RNAscope, the left and right NGs from 4-6 mice were dissected and analysed, while for the co-expression analysis, NGs from three mice were used. This information has now been added to the corresponding figure Legends 2, p. 6, line 206; and 3 p.8, line 269.

For tracer injection experiments, a total of 39 WT C57Bl/6j mice were used, with 3-5 mice allocated per each GIT region. This information is now included in the Material and Methods, p. 16, line 543 and P.16, line 558.

4.4. Immunohistochemistry

- how many sections were stained to determine the number of GLP1R neurons?

Author reply: As mentioned above, 4-11 cryosections per ganglion were collected and this information in the Material and Methods, p. 17, line 575.

- how many and how were the sections counted for the presence of traced neurons, how many animals were used to perform this study?

Author reply: As mentioned above, for tracer injection experiments, a total of 39 WT C57Bl/6j mice were used, with 3-5 mice allocated per each intestinal region. This information is now included in the Material and Methods, p. 16, line 543 and P.16, line 558. On average, 7 sections from each ganglion were analyse. The number of tracer positive neurons was normalized to the total number of NG neurons labelled with HuC/HuD. This is now mentioned in the Material and Methods, p. 17, line 570-573.

4.5. Stereology

How many mice were used for this study?

Author reply: As mentioned above, for stereological quantification, nodose ganglia from seven WT C57Bl/6j mice were used, see Material and Methods, p. 17, line 597, and Results section, p. 3, line 122.

4.9. Nodose ganglia explants preparation and field nerve potential recording

How many mice were used for this study?

Author reply: For field nerve potential recordings, six mice were used, as now indicated in the Material and Methods, p. 18, line 643. For Ca2+ imaging, NG were isolated from three mice, as now stated on page 18, line, 656.

References

All literature items should be prepared in accordance with the rules required by the journal.

Line 132 - after the literature item number 18 appears in the text number 27 instead of number 19 in the order

no citation of references 19 to 26 in the manuscript text

there is no order of numbering of cited literature items in the manuscript text, this should be sorted out as a whole

Author reply: We appreciate the reviewer for highlighting this important point. All references have now been properly organized.

Reviewer 3 Report

Comments and Suggestions for Authors

The abstract is difficult to follow. Please rewrite and present in a more organized way the aims and results.

Terms such as  'RNAscope in situ hybridization technology' must be explained briefly in the text.

Materials and Methods should appear before the Resuts section. Please correct the order of the section.

Figures are difficult to follow. Please describe in the main text of the Results.

How many and with what characteristics were the mice used? Please add more information describing the sample.

Authors should think about the small sample size and refer in the title that these findings are preliminary. In addition, multilevel analyses as in social sciences could be used (https://pmc.ncbi.nlm.nih.gov/articles/PMC9036373/   and   https://www.taylorfrancis.com/books/mono/10.4324/9781315650982/multilevel-analysis-joop-hox-mirjam-moerbeek-rens-van-de-schoot) as an alternative to classsical statistical analyses. This point can be mentioned in the proposal for future studies.

Authors need to discuss the clinical applications-dimensions of their findings to humans.

Author Response

The abstract is difficult to follow. Please rewrite and present in a more organized way the aims and results.

Author reply: Following reviewer’s suggestion, we have revised the Abstract to present the aims and results more clearly. However, we cannot explicitly separate it into 'Aims,' 'Results,' and 'Conclusion' sections, as this does not align with the journal’s formatting guidelines.

Terms such as  'RNAscope in situ hybridization technology' must be explained briefly in the text.

Author reply: We thank the reviewer for this valuable suggestion. A brief description of the methodology has now been added to the text, p. 3, line 105.

Materials and Methods should appear before the Resuts section. Please correct the order of the section.

Author reply: We sincerely appreciate your suggestion regarding the replacement of the Materials and Methods section. However, we must follow the journal's formatting guidelines, which specify that the Materials and Methods section should appear after the Results and Discussion sections. As authors, we are required to adhere to the journal's structure to ensure consistency across all published articles. We hope for your understanding in this matter and appreciate your valuable feedback, which has helped us improve our manuscript.

Figures are difficult to follow. Please describe in the main text of the Results.

Author reply: We apologize for the inconvenience. We have standardized figure references throughout the text, using “Figure” instead of “Fig.”. Additionally, we now explicitly refer to figures in the Results section, for example as “As shown in Figure 1c” (page 3, line 129). Additionally, we have revised the Venn diagram in Figure 3d to improve the clarity of the presented relationships and extended the Legend for Figure 3 to clarify what the Venn diagram represents.

How many and with what characteristics were the mice used? Please add more information describing the sample.

Author reply: In total, seven Glp1r-Cre/tdTomato-flox transgenic mice and 66 WT C57Bl/6j mice were used. This information is now included in the Material and Methods, p. 16, lines 542-544. Additionally, we have specified the exact number of animals used for stereology, IHC, and tracer experiments in their respective methodological sections.

Authors should think about the small sample size and refer in the title that these findings are preliminary. In addition, multilevel analyses as in social sciences could be used (https://pmc.ncbi.nlm.nih.gov/articles/PMC9036373/   and   https://www.taylorfrancis.com/books/mono/10.4324/9781315650982/multilevel-analysis-joop-hox-mirjam-moerbeek-rens-van-de-schoot) as an alternative to classsical statistical analyses. This point can be mentioned in the proposal for future studies.

Author reply: We thank the reviewer for this valuable suggestion. The standard sample size in animal research, particularly when using mice, varies depending on the type of study, expected effect size, variability in the data, and ethical considerations. Ethical committees require a power analysis to justify sample size, and we are not permitted to exceed the approved number of animals. Howevere, typical sample size for experiemmnts in mice is as folow: behavioral studies: 8–12 mice/group; molecular/biochemical studies: 5–10 mice/group; histological/(ICH) studies: 3–6 mice/group; pharmacological/intervention studies: 6–12 mice/group; and genetic studies: 10–20 mice/group. In most of our experiemnts, we used seven mice per group, which falls within these standartd ranges. Nevertheless, we acknowledge the reviewer's concern and have now explicitly indicated in the manuscript that our findings are preliminary, see p. 3, line 114. Additionally, we have noted that the tracer experiments showed high variability; however, since our results align with previous findings reported by other researchers, we decided not to increase the number of animals, in accordance with the 3Rs principles (Replacement, Reduction, and Refinement).

Regarding the suggestion to apply multilevel analysis, we appreciate the reviewer’s input but believe this approach may not be appropriate for our histological and molecular data. Multilevel models are particularly valuable in social sciences and similar fields where data are structured hierarchically (e.g., individuals nested within social groups or organizations). These models help account for group-level influences on individual-level outcomes.

In contrast, our study treats each animal as an independent biological unit, with measurements collected at the individual level rather than within a nested or hierarchical structure. Applying multilevel analysis in this context could introduce unnecessary complexity without adding meaningful insights. Furthermore, improper use of multilevel modeling can lead to loss of statistical power due to aggregation or inflated significance due to disaggregation, potentially compromising the validity of our results.

For these reasons, we have opted for statistical approaches that align more directly with the structure of our data, ensuring clarity and robustness in our findings. We appreciate the opportunity to address this point and would be happy to discuss it further if needed.

Authors need to discuss the clinical applications-dimensions of their findings to humans.

Author reply: We appreciate the reviewer’s suggestion to discuss the potential clinical implications of our findings. However, as mentioned earlier, our data are preliminary and represent an early stage of research, far from any direct application to human or clinical settings. At this point, our study primarily aims to explore fundamental biological mechanisms rather than to establish translational relevance. While we acknowledge the broader interest in potential clinical applications, drawing direct conclusions about human relevance based on our current findings would be premature and speculative. Further research, including validation in more complex models and additional preclinical studies, would be necessary before considering any translational implications. For these reasons, we believe that the discussion of potential clinical applications is beyond the scope of our study. However, we recognize the importance of this aspect and may explore these connections in future research.

Round 2

Reviewer 1 Report

Comments and Suggestions for Authors

The comments and suggestions from the previous review have been adequately addressed. Accordingly, the manuscript is now considered suitable for publication in the journal.

Author Response

Thank you for your comments. We appreciate your positive feedback.

Reviewer 2 Report

Comments and Suggestions for Authors

Dear Authors, Thank you very much for sending all the information I requested and for completing it in the manuscript text, and I accept the manuscript for publication in its current form.

Author Response

(The authors gave the same response as above.)
